# GJB2-Related Hearing Loss: Genotype-Phenotype Correlations, Natural History, and Emerging Therapeutic Strategies

**DOI:** 10.3390/ijms27010491

**Published:** 2026-01-03

**Authors:** Julia Anne Morris, Tomas Gonzalez, Susan H. Blanton, Simon Ignacio Angeli, Xue Zhong Liu

**Affiliations:** 1Department of Otolaryngology, University of Miami Miller School of Medicine, Miami, FL 33136, USA; jam730@miami.edu (J.A.M.);; 2Morsani College of Medicine, University of South Florida, Tampa, FL 33602, USA; 3Dr. John T. Macdonald Foundation Department of Human Genetics, University of Miami Miller School of Medicine, Miami, FL 33136, USA; 4Hussman Institute for Human Genomics, University of Miami Miller School of Medicine, Miami, FL 33136, USA; 5Department of Pediatrics, University of Miami Miller School of Medicine, Miami, FL 33136, USA

**Keywords:** *GJB2*, connexin-26, genetics, non-syndromic hearing loss, natural history, genotype-phenotype correlations, congenital hearing loss, gene therapy, cochlear implant outcomes

## Abstract

This review integrates molecular, clinical, and translational data to provide an updated understanding of *GJB2*-related deafness and its emerging treatment landscape. Truncating mutations in *GJB2* typically cause severe-profound hearing loss (HL) phenotypes, whereas non-truncating alleles are often associated with milder or progressive phenotypes. Geographic variation in variant prevalence contributes to regional differences in disease burden. Beyond the coding region, deletions and cis-regulatory mutations within the DFNB1 locus, including *GJB6* and *CRYL1*, can influence HL severity when compounded with other pathogenic *GJB2* variants. DFNB1 hearing loss generally presents as symmetric, bilateral, and flat to gently sloping across frequencies, with preserved cochlear neurons that support excellent cochlear implant (CI) outcomes. Early implantation CI in *GJB2*-positive children yields superior speech and language development compared with non-*GJB2* etiologies. Emerging therapies include dual-AAV (AAV1 + AAV-ie/ScPro) delivery, achieving cell-specific Cx26 restoration, adenine base-editing for dominant-negative variants, and allele-specific suppression using RNA interference or antisense oligonucleotides. Concurrent progress in human iPSC-derived cochlear organoids provides a physiologic model to advance toward clinical trials. By integrating genotype-phenotype correlations, natural history insights, and advances in molecular therapeutics, this review presents a comprehensive update on *GJB2*-related HL and highlights how gene-based strategies are poised to change the treatment of this condition.

## 1. Introduction

Hearing is a complex perceptual sense that enables humans to communicate, interpret environmental cues, and form social and emotional connections. Thus, hearing loss can have a profound impact on the cognitive, psychosocial, and educational development of an individual throughout life [1]. Functional hearing relies on a coordinated multistep process, known as mechanotransduction, in which sound waves are funneled through the outer ear, vibrate the tympanic membrane, and are transmitted via the ossicular chain into the fluid within the cochlea. Within the cochlea, specialized hair cells lining the organ of Corti transduce these waves into neural signals that are then sent to the brain [1,2,3].

Given this complexity, it is unsurprising that approximately 1% of the human coding genome is involved in the development and function of the auditory system [3,4]. Hearing impairment is the most common sensory disorder, affecting around 1 in 500 newborns worldwide [5], and the etiology of hearing loss is highly heterogeneous, with genetic and environmental causes contributing to its high prevalence. Genetic causes account for approximately 80% of pre-lingual hearing loss in developed countries, and, to date, over 125 genes have been implicated in non-syndromic hearing loss (NSHL) [6]. Additional contributions from syndromic genes, genetic modifiers, and non-coding regions further underscore [3] the heterogeneous and complex mechanism of hereditary hearing loss.

Historically, genetic hearing loss has been classified according to its inheritance pattern, with DFNA denoting dominant forms, DFNB recessive, and DFNX X-linked. Among these, the DFNB1 locus on chromosome 13q11–12, represents the most common cause of autosomal recessive NSHL, accounting for up to 50% of cases, although this varies by population [3,6,7]. Within this locus, mutations in the *GJB2* gene alone account for more than half of all molecularly confirmed HL diagnoses worldwide [6].

The *GJB2* gene (GenBank M86849, MIM 121011) encodes connexin 26 (Cx26), a gap junction protein that was first identified as a deafness gene in 1997 [8]. Connexin 26 is a critical component of the cochlear gap junction network and is abundantly expressed in both the cochlear-epithelial cells and the fibrocyte supporting cells of the cochlea, where it plays a major role in potassium ion (K^+)^ recycling within the endolymph [9]. K+ circulation is essential for maintaining the electrochemical gradient responsible for generating action potentials in response to sound. Specifically, Cx26 forms hexameric hemichannels, also known as connexons, that dock with adjacent connexons to assemble functional gap junction channels, enabling intercellular diffusion of ions, metabolites, and signaling molecules. In the cochlea, Cx26 co-assembles with connexin 30 (Cx30), encoded by *GJB6*, to form heteromeric gap junctions primarily in supporting cells and fibrocytes. This network facilitates the recycling of K+ ions following mechanotransduction. Upon sound stimulation, mechanical deflection of hair cell stereocilia causes depolarization and K+ influx from the K+-rich endolymph of the scala media, after which K+ effluxes basolaterally into the normally K+-poor perilymph and is recycled back to the endolymph via these interconnected supporting cells, fibrocytes in the spiral ligament, and the stria vascularis, preventing toxic buildup and supporting the electrochemical gradient required for mechanotransduction. In addition to ion homeostasis, Cx26 also participates in intracellular and intercellular signaling critical for cochlear development and function, including ATP release and calcium signal wave propagation [9,10]. Disruptions in these processes by *GJB2* variants lead to impaired cochlear homeostasis and sensorineural hearing loss.

Much of the current understanding of Cx26 physiology has been derived from mouse models. Complete *GJB2* knockout is embryonically lethal due to impaired placental glucose transport, necessitating the development of conditional and inducible mouse models to study auditory phenotypes. Cochlea-specific deletion of *GJB2*, particularly within supporting cell populations, disrupts gap junction networks, impairs potassium recycling and intercellular signaling, arrests normal organ of Corti maturation, and results in sensorineural hearing loss. These mouse models have been instrumental in defining the physiological role of Cx26 in mammalian hearing and have served as in vivo platforms for testing emerging gene therapy and genome-editing strategies for *GJB2*-related deafness [9,11].

Despite significant advancements in understanding *GJB2*-associated Cx26 hearing loss, challenges remain. The condition is highly heterogeneous, even among individuals with the same genotype, with broad variation in onset, severity, and progression [12,13]. This variability complicates prognosis, genotype-phenotype interpretation, and genetic counseling. Clarifying the molecular pathophysiology and phenotypic spectrum of *GJB2*-related hearing loss is essential to define the natural history of *GJB2* deafness and lay the clinical foundation for future gene, cell, and drug-based therapeutics.

## 2. Materials and Methods

We conducted a PubMed search (1997–2025) using the keywords: “*GJB2*”, “DFNB1”, “Connexin 26”, “genotype-phenotype”, “progression”, “audiology”, “cochlear implant”, “imaging”, “*GJB6*”, “*CRYL1*”, “non-coding”, “AAV”, “gene therapy”, “organoid”, “base editing”, “truncating”, and “non-truncating”. Articles were included if they provided data on clinical phenotypes, genotype–phenotype associations, non-coding variants, imaging, or emerging therapeutics for *GJB2*-related hearing loss. Peer-reviewed English-language publications were prioritized; selected preprints were included only when directly relevant to emerging therapeutic strategies.

## 3. Molecular Basis of *GJB2* Hearing Loss

The *GJB2* gene consists of a 160-bp non-coding exon, a single 3,179-bp intron, and a 2,134-bp coding exon containing a 22-bp 5′-untranslated region (UTR), the full 678-bp coding sequence, and the 3′-UTR (UniGene Hs.524894) [14]. Pathogenic variants (PVs) in *GJB2* represent the most common cause of congenital, autosomal recessive non-syndromic sensorineural hearing loss (NSHL) [15]. More than 100 distinct PVs have been reported, nearly all within the coding region, making *GJB2* one of the most frequently interrogated genes on targeted next-generation sequencing panels (NGS), whole-exome sequencing (WES), and other kinds of hearing loss assays in clinical practice [16,17,18].

PVs can be broadly classified into two functional categories: (i.) truncating variants, including frameshift, nonsense, or splice-altering mutations, that introduce premature termination and typically result in Cx26 loss of function, and (ii.) non-truncating variants, primarily missense mutations, that preserve protein length, but impair protein activity. Common pathogenic variants in both functional categories, along with their molecular consequences, associated phenotypes, and population origins, are summarized in Table 1; structural localization shown in Figure 1 and geographic origin illustrated in Figure 2.

### 3.1. Truncating Mutations

The c.35delG (p.Gly12fs) frameshift is the most prevalent GJB2 mutation worldwide and is particularly common in populations of European ancestry, where it strongly correlates with congenital, profound NSHL [19,20,21,22,23,24]. Haplotype analysis suggests that the origin of this mutation arose from a founder effect in ancient Turkish/Anatolian populations, consistent with its high prevalence across European and Middle Eastern populations [24]. c.35delG introduces a premature stop codon in the second exon, producing a nonfunctional Cx26 protein [20].

Several other truncating PVs show clear population specificity and defined molecular consequences. The c.235delC (p.Leu79fs) frameshift causes a subsequent termination in codon 81. It was first reported in a Japanese cohort and predominates in East Asian populations [21,25,26,27,28]. Similarly, c.71G>A (p.Trp24Ter; W24X) is very common in South Asia, with the highest prevalence in India, and results in a truncated protein that is retained intracellularly rather than trafficked to the membrane [20,29,30,31]. The c.167delT (p.Leu56fs) allele is a single base pair deletion, prevalent in the Israeli and Ashkenazi Jewish populations, with some reports in the Iraqi population [20,32,33].

Although less common, insertion and duplication variants are another etiology of truncating mutations. A 2024 report from the Ivory Coast described c.205_21dupTTCCCCA (p.Ser72ProfsTer32) in a child with severe congenital NSHL, identified in compound with variant c.132G>A (p.Trp44Ter) [34]. This duplicating variant was classified as pathogenic by the Moroccan Genomic and Human Genetics Laboratory in 2024 based on the 2015 ACMG criteria [34,35]. Notably, p.Trp44Ter (c.131G>A/c.132G>A; W44X) represents a founder allele in a Guatemalan cohort of Mayan ancestry, accounting for 21 of 266 pathogenic alleles [36]. Another high-impact truncating variant, c.605ins46, introduces a 46-base-pair insertion causing frameshift and early termination [37]. c.605ins46 has been found in both homozygotes and compound heterozygotes in pre-lingual NSHL patients across Asian cohorts [37,38,39,40].

Population studies underscore how regional ancestry shapes modern genetic diversity of *GJB2* truncating alleles. In a South Florida cohort, the frequency of pathogenic *GJB2* alleles was comparable between White (35%) and mixed-Hispanic (34%) individuals; however, variant frequency differed substantially. White probands carried 33% of identified truncating PVs, whereas mixed-Hispanic probands carried 83%. Only c.35delG and c.167delT were shared across both groups. This contrasts with earlier and later reports from Mexico and the Caribbean, where truncating GJB2 PVs were found to be relatively uncommon [41,42]. Mixed-Hispanic individuals were nearly three times more likely to harbor a truncating variant other than c.35delG, emphasizing how regional diversity can contribute to substantially greater DFNB1 genetic diversity [43].

### 3.2. Non-Truncating Mutations

Non-truncating PVs are generally less common than truncating alleles but still show clear population clustering [44,45]. The c.427C>T (p.Arg143Trp) missense variant results in abnormal protein function [34]. It is particularly prevalent in West African populations, with evidence that it originated in Ghana [34,46,47]. Bioinformatic analysis supports its deleterious impact on Cx26 function [48]. Missense variants cluster within functionally critical domains of Cx26, including the extracellular loops required for connexon docking and channel gating (Figure 1), whose tertiary structure has been defined through biochemical and structural studies [49,50].

The c.101T>C (p.Met34Thr) variant is one of the earliest described missense *GJB2* alleles and has undergone extensive evaluation [8]. Current evidence supports its pathogenicity, particularly when detected in trans with a truncating variant [4,28,51,52,53]. M34T is prevalent in European populations and typically produces mild to moderate NSHL [54].

The c.109G>A (p.Val37Ile) missense variant, first described in a Kenyan patient in 1998, is now recognized as a common allele in East Asia and Southeast Asian populations [51,55]. Functional studies indicate that this variant causes a marked negative effect on the functional activity of Cx26, and predictive in silico tools consistently classify this variant as disease-causing [56,57].

The c.257C>G (p.Thr86Arg) substitution, documented in Chinese, Japanese, and Korean cohorts, is characterized by a change in amino acid charge from neutral to positive and results in impaired channel oligomerization and aberrant localization to the cytoplasm in a reticular pattern, correlating with moderate hearing loss [39,58].

Other population-specific non-truncating *GJB2* variants include P.Trp172Cys (516G>C), which likely originated in Southern Siberia, accounting for 62.9% of PVs and 3.8% carrier frequency in a Tuvinian cohort [59]. Variant p.Leu90Pro (c.269T>C) is another missense mutation found in many populations globally, resulting in mild to moderate hearing impairment through partial disruption of channel activity [18].

The c.379C>T (p.Arg127Cys) variant shows variable clinical severity and ambiguous pathogenicity in studies from 2001 and 2006 [60,61]. It has now been classified as pathogenic/likely pathogenic due to findings in compound heterozygotes, large population studies, and in silico predictive models [45,48,61,62]. Currently, no ethic or geographic origin has been established.

A subset of *GJB2* missense variants cause autosomal dominant HL through dominant-negative or gain-of-function mechanisms. Most cluster on the first extracellular loop of GJB2 (E1), which is essential for connexon-connexon docking [18]. The p.Arg75Trp (c.223C>T; R75W) and p.Arg75Gln (c.224G>A; R75Q) variants impair gap junction formation and cause severe prelingual HL with variably penetrant Palmoplantar Keratoderma [63]. The p.Gly45Glu (c.134G>A) variant is an established cause of keratitis-ichthyosis-deafness (KID), producing constitutively open hemichannels and profound early onset HL [64].

**Table 1 ijms-27-00491-t001:** Common Truncating and Non-Truncating *GJB2* Mutations with Molecular Consequences, Phenotype, and Variant Origin.

Mutation Type	Protein Change	Chromosomal Change	Molecular Consequence	Phenotype	Origin	Citation
Truncating	p.Gly12fs	c.35delG	Frameshift	Severe to profound bilateral congenital NSHL	Europe	[23,24,65]
	p.Leu79fs	c.235delC	Frameshift	Severe to profound bilateral congenital NSHL, with some cases of asymmetric NSHL	China	[26,27,66]
	p.Trp24Ter	c.71G>A	Termination	Variable, Mild to moderate bilateral congenital NSHL	South Asia	[54]
	p.Leu56fs	c.167delT	Frameshift	Severe to profound bilateral congenital NSHL	Eurasia	[44,67]
	p.Trp44Ter	c.131G>A	Termination	Severe to profound bilateral congenital NSHL	Guatemala	[36]
	p.Ser72ProfsTer32	c.205_21dupTTCCCCA	Termination	Severe bilateral congenital NSHL	West Africa	[35]
	-	c.605ins46	Termination	Severe to profound bilateral congenital NSHL	Japan	[37]
Non-Truncating	p.Arg143Trp	c.427C>T	Missense	Moderate to profound bilateral congenital NSHL	Ghana	[46,47]
	p.Met34Thr	c.101T>C	Missense	Mild to moderate bilateral congenital NSHL	Europe	[54]
	p.Val37Ile	c.109G>A	Missense	Mild to moderate bilateral congenital NSHL	East Asia	[55]
	p.Thr86Arg	c.257C>G	Missense	Moderate to profound bilateral congenital NSHL	East Asia	[37,68]
	p.Arg127Cys	c.379C>T	Missense	Moderate to severe bilateral congenital NSHL	Undetermined	[45]
	p.Trp172Cys	c.516G>C	Missense	Mild to profound bilateral congenital NSHL	Eastern Europe	[69]
	p.Leu90Pro	c.269T>C	Missense	Mild to moderate bilateral congenital NSHL	Middle East	[7,70]

**Figure 1 ijms-27-00491-f001:**
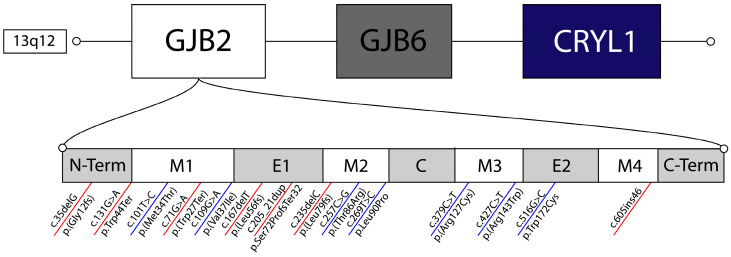
Structural localization of truncating (red) and missense (blue) *GJB2* variants within chromosome 13q12, including nearby *CRYL1* and *GJB6* regions. The bottom panel provides a simplified linear schematic of Cx26 topology; for detailed 3D protein structures illustrating variant impacts on connexon formation, see [49,50]. E1/E2: extracellular loops; M1–M4: transmembrane domains; C: cytoplasmic loop.

**Figure 2 ijms-27-00491-f002:**
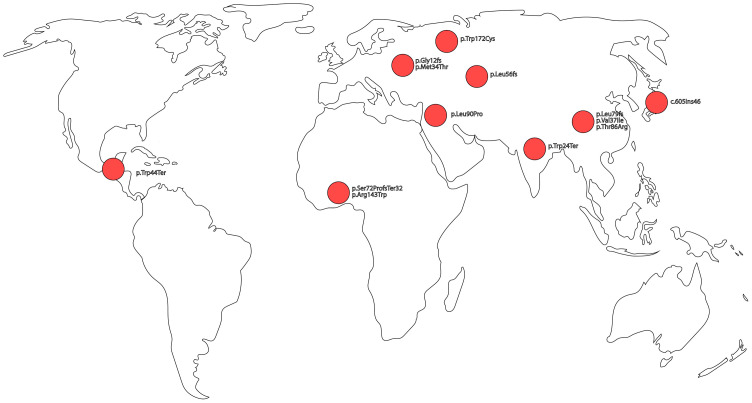
Suspected geographic origin of pathogenic *GJB2* variants associated with autosomal recessive non-syndromic hearing loss (DFNB1).

## 4. The DFNB1 Locus Beyond the Coding Sequence

The high prevalence of *GJB2* mutations across diverse populations has made this gene an essential part of molecular testing for hereditary hearing loss. These variants are typically inherited in an autosomal recessive manner, resulting in DFNB1-related hearing loss in individuals who are homozygotes or compound heterozygotes for a pathogenic allele [8,71]. However, several studies have reported that 10–50% of symptomatic individuals possess a single heterozygous truncating *GJB2* allele, which is generally presumed to function as a null allele and would not be expected to cause hearing loss in isolation. This observation complicates molecular diagnosis and suggests that additional pathogenic variants may exist within the DFNB1 locus but outside the *GJB2* coding region [72,73].

Mutations within the DFNB1 locus can be broadly classified as either: (i.) those that directly alter the *GJB2* coding sequence, and (ii.) those that disrupt its regulation or expression through effects on non-coding regions [74]. The latter class of variants, found in promoters, introns, or intergenic regions, is often missed by routine genetic testing, such as targeted sequencing panels and whole exon sequencing (WES), which generally exclude non-coding regions. Thus, detection requires whole genome sequencing (WGS), which is more comprehensive but resource-intensive [75,76].

Beyond *GJB2*, several neighboring genes within the DFNB1 locus influence its expression and phenotype. Large genomic deletions that disrupt regulatory regions or adjacent genes can inhibit *GJB2* expression while leaving its coding sequence intact [73,77]. The first reported deletions were del (*GJB6*-D13S1830) and del (*GJB6*-D13S1854), spanning 309 kb and 232 kb, respectively. Both remove segments of *GJB6* and the nearby *CRYL1* gene, which includes the first five exons of *GJB6*, and the whole *CRYL1* gene, that removes a 232-kb interval, with the proximal breakpoint within *GJB6* intron 5, and the distal breakpoint in intron 4 of *CRYL1* [78,79,80,81,82].

Additional deletions have since been documented including: (i.) a 920kb deletion of the *PSPC1, CRYL1, ZMYM5, ZMYM2, MPHOSPH8, GJA3*, *GJB2*, and *GJB6* genes in a prelingual HL patient, (ii.) del (*GJB2*-D13S175), a 101kb deletion observed a multi-ethnic Russian cohort, (iii.) a 131.4kb deletion identified in a German-American family with congenital non-syndromic HL, (iv.) a 95kb deletion detected in a multi-ethnic cohort, and (v.) an 8kb deletion reported in a Japanese patient with profound HL [73,83,84,85,86].

The *CRYL* region has gained recent attention as a regulatory hub for *GJB2*. The *CRYL1* gene is located on chromosome 13q12.11 and sits ~200kb away from *GJB2* and contains cis-regulatory elements essential for *GJB2* expression [87]. Thus, even in patients with one *GJB2* pathogenic allele, if an additional deletion in *CRYL1* exists, autosomal dominant SNHL can present [88]. A variant in *CRYL1*, namely c.261T>C (p.Gly87Gly), was shown to have a positive correlation to more severe HL phenotypes for patients with a homozygous variant c.109G>A (p.Val37Ile) in *GJB2* [13].

The *GJB6* gene encodes Connexin 30 (Cx30), located ~ 35 kb telomeric to *GJB2.* Initially identified in autosomal dominant deafness (DFNA3) *GJB6* mutations were later shown to participate in digenic inheritance with *GJB2* [79,82]. Functional studies in humans and rodents demonstrated that Cx26 and Cx30 share the same spatial pattern of expression within the cochlea, with apparent co-localization, and can form heteromeric connexons and heterotypic gap-junction channels [61,77,89,90]. Mutant Cx26 subunits involved in autosomal dominant NSHL can exert dominant negative effects on wild-type Cx30 [90,91]. In murine models, *Gjb6* knock-out (*Gjb6*^tm1Kwi/tm1Kwi^) results in severe deafness and absent endocochlear potential due to disruption of the endothelial barrier of capillaries embedded within the stria vascularis, confirming its essential role in cochlear ion homeostasis [92,93].

Further, noncoding variants within *GJB2*’s promoter and intronic regions have gained attention for their pathogenic potential. Using WGS, Nabec et al. 2021 [94] identified a promoter variant (c.139G>A; p, Glu47Ter), at the GC box –81 bp of the transcription start site, responsible for basal transcription of *GJB2* [95]. Although currently listed in ClinVar with conflicting pathogenicity classifications due to limited evidence.

Another splice-site variant, c.-23+1G>A, located in intron 1 at the splice donor site, has been reported in multiple Eurasian populations including Tuvian, Yakut, and Mongolian [59,96]. ClinVar lists this variant as Pathogenic/Likely Pathogenic based on current submissions. Similarly, Kashef et al., 2015 identified 17 Iranian probands carrying IVS1+1G>A in exon 1, now classified as pathogenic/likely pathogenic in ClinVar [97]. These findings further reinforce that population-specific non-coding variants are widely prevalent but can be missed by current clinical sequencing methods.

Collectively, research throughout the past two decades demonstrates that DFNB1-related HL is not simply caused by variations in the *GJB2* coding region. Large structural deletions, *GJB6* digenic inheritance, *CRYL-1*-mediated regulation, and promoter and splice variants all contribute to the heterogeneity of this disorder. As current clinical diagnosis relies on WES and targeted panels, which omit these regions, WGS should be utilized to further develop our understanding of the DFNB1 locus and improve diagnostic accuracy.

## 5. Clinical Manifestations and Natural History

DFNB1-related hearing loss (HL) is classically bilateral and prelingual, though age of onset and severity vary considerably by genotype [70]. Apart from hearing loss, those with this condition typically experience normal development and vestibular function [71]. Those carrying two copies of truncating variants typically experience early onset, severe to profound HL, whereas individuals with at least one non-truncating (missense) variant often present with mild-to-moderate or progressive HL due to residual Cx26 function [13]

Vestibular dysfunction and temporal bone malformations are uncommon, which is to be expected as *GJB2* expression has not been found in the temporal bone or vestibular nerve. Some rare cases have reported findings of dilated endolymphatic fossae, hypoplastic modiolus, large vestibular aqueducts, hypoplastic horizontal semicircular canals, and hypoplastic cochlea on imaging [71,98]. Kenna et al., 2011 found that in a 113-patient cohort, around 10% of *GJB2* patients had inner ear abnormalities, but most were subtle and showed no correlation with HL severity [44].

Although DFNB1-related HL is often congenital, delayed onset and progressive phenotypes have been recognized. In a Japanese cohort, 6.9% of children with biallelic *GJB2* variants passed the newborn hearing screen but developed HL later, underscoring the need for continued audiologic surveillance and genetic follow-up after newborn screening [99]. Longitudinal data from Sakata et al., 2022 further demonstrated that HL progression can occur across genotypes, particularly in compound heterozygotes carrying one truncating and one missense allele [45].

With the accumulation of longitudinal data, our understanding of the progression of *GJB2* has further developed. Kenna et al., (2010) found that in patients with residual hearing, over 50% experienced a gradual or precipitous progression of their HL [44]. Progression tends to be gradual and affects the mid-high frequencies first and is more common in missense/non-truncating or compound genotypes where Cx26 remains partly functional [45]. The systematic review by Chan et al., 2014 showed that in 28 of the 216 articles, 18.4% of patients showed progression [21]. A long-term observation study by Sakata et al., 2022 identified progression of HL in 6.5% of patients [45]. They found p.Leu79Cysfs*3 variants correlated with a progressive phenotype. Additionally, they emphasized that those with progression were observed initially with moderate to severe hearing loss [45].

Chiang et al., 2023 further demonstrated that even among c.109G>A (p.Val37Ile) homozygotes, a subset of patients showed progressive decline, which supports the hypothesis that environmental exposures and genetic modifiers are important to determine HL trajectory [13]. Recognizing this variability is crucial for counseling families and determining the timing of intervention.

### 5.1. Audiology

Audiometric profiles of DFNB1 HL are typically bilateral, symmetric, and characterized by flat or gently down-sloping across frequencies [70]. Severity spans the full range from mild to profound, with most individuals with DFNB1 HL presenting with severe to profound impairment [21]. Severity often correlates with genotype, as truncating mutations are more frequently associated with more severe hearing deficits than missense or non-truncating alleles [13,44]. In a combined cohort of 234 individuals with biallelic *GJB2* mutations, mild (4.5%) and moderate (13.3%) hearing loss accounted for only a small fraction of cases, whereas 14.9% exhibited severe hearing loss, and the majority (25%) had profound deafness. These data demonstrated that the probability of identifying biallelic pathogenic *GJB2* variants increases with greater audiologic severity, although individuals with mild to moderate hearing loss should not be excluded from genetic testing [100].

### 5.2. Cochlear Implant Outcomes

Children with *GJB2*-related severe to profound HL generally achieve great cochlear implant outcomes, reflecting preservation of the cochlear architecture and the auditory nerve. As *GJB2* affects the supporting cells of the inner ear, rather than the cochlear neural circuitry, cochlear implant (CI) mediated electrical stimulation can bypass the deficit in the supporting cells and effectively restore auditory function [70].

Wu et al., 2015 evaluated long-term CI outcomes in *GJB2*-associated hearing loss, using measures of auditory perception and speech development over a 3–5 year follow-up period [101]. Their results showed that *GJB2*-positive patients showed significant gains in speech and language development to a greater degree than those with non-*GJB2* deafness. Notably, early implantation was associated with the most improvement, and later implantation with worse outcomes.

A comprehensive meta-analysis by Nishio & Usami 2017, encompassing more than 35 studies on *GJB2* cochlear implant outcomes, further confirmed these findings [102]. Across cohorts, GJB2 mutation carriers consistently exhibited equal or superior performance compared with other HL etiologies, and further studies have shown that even when controlling for residual hearing, age at implantation, and duration of implant use, DFNB1 children who use CIs exhibit greater gains in expressive language than non-DFNB1 children [103]. Though one study by Lalwani et al., 2009 showed that later implantation of CI in GJB2 patients showed worse outcomes, and increased device usage was associated with better scores [104]. These findings further underscore the importance of early implantation for achieving optimal CI outcomes in GJB2 patients.

## 6. Genotype-Phenotype Correlations

Large multicenter analyses have shown that the severity of hearing loss among individuals with *GJB2* variants is highly variable, even for the same genotype. In a cohort of over 1500 subjects across 16 countries Snoeck et al., 2005 and later Chiang et al., 2023 demonstrated that hearing loss severity ranged from mild to profound, but when grouping by the molecular effect, Truncating (T) vs. Non-truncating (NT), patterns emerged [7,13].

Genotypes containing two non-truncating variants (NT/NT) generally exhibit less severe HL than those with two truncating variants (T/T) [7,45]. T/T genotypes typically produce severe to profound hearing loss, with a minority of cases having mild to moderate hearing loss (12% and 3% respectively) [45]. Compound heterozygotes (T/NT) demonstrate intermediate severity HL, with around half of these cases having severe to profound hearing loss [44]. For example, c.109G>A (p.Val37Ile) (NT) is commonly associated with mild to moderate NSHL, whereas patients with c.235del (p.Leu79fs) (T), typically experience profound HL [105]. This difference can be attributed to the severe structural and functional changes caused by a truncating mutation, when compared to a non-truncating mutation.

Certain genotypes display reproducible audiometric phenotypes. Most cases with genotypes c.35delG/p. (Arg143Trp) or c.35delG/del (*GJB6*-D13S1830) show profound HL, while mild or moderate HL is more frequent in subjects carrying p. (Leu90Pro), p. (Met34Thr), or p. (Val37Ile) in combination with a truncating variant [7]. These conclusions were confirmed by further studies across multiple cohorts [16,43,44,106].

The missense/NT variant p.Val37Ile is described as a hypomorphic allele with high carrier frequency and is typically associated with mild-moderate HL. However, considerable variation in phenotypes exists. In a Taiwanese cohort of individuals homozygous for this variant, 35 patients presented with severe-to-profound HL, and in 25.3% of these cases, pathogenic variants in additional deafness-associated genes (*SLC26A4*, *TECTA*, and *SIX1*) were identified. Further, a cis-regulatory modifier found upstream to *GJB2* (*CRYL1 rs14236*) was significantly associated with more severe hearing thresholds among p.Val37Ile homozygotes and is hypothesized to enhance expression of the p.Val37Ile variant allele. Together, these results emphasize that the severity of DFNB1 HL cannot be predicted by *GJB2* genotype alone and that modifier genes and regulatory elements contribute to significant variability in phenotype [13].

The wide phenotypic spectrum of DFNB1 HL likely reflects, at least in part, the influence of genetic modifiers. A large whole-genome association study investigating the phenotypic variability of c.35delG homozygotes found that the variability in hearing severity is not caused by one major genetic modifier, but instead by a set of nine-single nucleotide polymorphisms (SNPs) each individually contributing small modifying effects [107]. In addition, Rasember et al., 2013, identified another SNP proximal to the *GJB2* transcription start site that may alter tissue and developmental stage regulation of *GJB2* gene expression [108]. Collectively, these studies highlight the regulatory complexity that exists within the DFNBB1 locus, which underscores the need for comprehensive genome analysis beyond the traditional *GJB2* region to better understand genotype-phenotype correlations.

## 7. Therapeutics and Future Directions

Over the past decade, major advancements in auditory genetics and vectorology have transformed the potential for treating genetic hearing loss, paving the way for precision interventions for *GJB2*-related deafness. Successful in vivo gene-therapy trials for OTOF-related deafness have demonstrated that restoring auditory function in humans is achievable, yet there are no current corrective therapeutic options for the treatment of *GJB2*-related deafness [109,110,111,112]. Ongoing clinical trials (NCT05901480, ChiCTR2200063181, NCT05821959, NCT05788536) report encouraging safety and efficacy data. These milestones provide a translational framework for treating *GJB2*-related hearing loss. Current research spans adeno-associated virus (AAV)-mediated gene replacement, allele-specific suppression, genome editing, and improved modeling systems, all of which aim to move beyond supportive devices to correct or bypass the auditory deficit at a molecular level.

### 7.1. Gene Therapy

*GJB2* has remained a difficult gene therapy target due to the diverse molecular mechanisms that cause *GJB2*-related hearing loss and the broad function and location of Cx26 across the cochlea [113]. Naturally occurring AAVs effectively transduce cochlear hair cells but have shown limited access to the supporting cells where Cx26 functions. The development of synthetic AAVs and cell-specific promoters has begun to overcome these barriers [113].

Studies using AAV-related methods for gene therapy showed initial success but were limited by vector design. In 2023, Jiang et al. showed robust rescue of hearing when AAVs were delivered perinatally in their mouse model, but diminished effects with later intervention (p42), emphasizing the importance of vector design and delivery timing [114].

In 2025, groundbreaking research by Sun et al., introduced a dual vector strategy by combining AAV-1 and AAV-ei carrying *GJB2* under the control of a supporting cell specific promoter ScPro. This approach prevented the ototoxicity seen with ectopic *GJB2* and achieved auditory recovery in conditional knockout mice. This approach in Bama miniature pigs and cynomolgus monkeys demonstrated safety through normal auditory thresholds and minimal systemic toxicity in the animals who received it. These findings mark the first in vivo demonstration of durable, cell-specific *GJB2* restoration and pave the way for future clinical applications [115].

Commentary by Landeggar 2025 emphasized that this combined AAV-1 AAV-ie/SCpro strategy represents a method that can target both epithelial supporting cells and the cochlear lateral wall, while avoiding ectopic Cx26 expression. Together, these studies illustrate how delivery strategy and vector design are crucial for successful *GJB2* replacement.

Adjunctive pharmacologic therapy may further optimize outcomes. Wang et al. Demonstrated that AAV-mediated gene therapy followed by low-dose Dexamethasone inhibited inflammatory responses in the inner ear, reduced outer hair cell loss, and improved hearing thresholds in mice [116].

Finally, in vivo chromatin profiling has identified gene regulatory elements (GRE) that allowed targeting of the supporting cell types and AAV delivery of HA-tagged *GJB2*. This method showed prevention of hair cell degradation and could potentially apply to hearing loss prevention in DFNB1 patients [117].

### 7.2. Allele-Specific Suppression

Allele-specific gene suppression offers an alternative therapeutic approach for dominant-negative *GJB2* variants, which require silencing rather than replacement. RNA interference (RNAi) approaches have successfully silenced pathogenic transcripts from variants, such as p.R75W, improving the function of connexon 26 [118]. However, RNAi methods are generally temporary, may require repeated administration, and pose a risk of off-site interactions [112]. Antisense oligonucleotide (ASO) technologies enable sequence specific transcript modulation, but their application to *GJB2*-related HL remains limited and requires further research [114].

### 7.3. Gene Editing

In regard to gene editing, CRISPR/Cas9 systems and related DNA editing systems are revolutionary tools for permanent correction of genetic defects. While current research has validated the efficacy of DNA editing systems for TMC1 and USH2A, the absence of *GJB2*-specific CRISPR studies highlights a critical research gap [119]. The 2025 study by Ukaji et al. demonstrated an AAV-mediated adenine-base-editing system that successfully repaired the dominant negative *GJB2* R75W mutation, characterized by a single incorrect base substitution. By using a compact SaCas9-NNG ABE8e packaged into a single AAV (AAV-Sia6e, the authors were able to achieve a 46.8% base conversion rate and reestablish functionality in both neonatal and adult transgenic mice. This achievement demonstrates the feasibility of genome editing for dominant *GJB2*-related conditions [118].

### 7.4. Model Systems and Translational Barriers

Therapeutic development is limited by a lack of in vivo models. Complete *GJB2* knockout is embryonically lethal in mice due to its essential role in placental glucose transfer, necessitating the use of conditional or mosaic mice models [114]. Moreover, the mature human cochlea is a much less permissive environment than the neonatal mouse, making gene delivery more effective in these models than we can expect in clinical applications.

To bridge this gap, human induced pluripotent stem cell (IPSC) derived otic organoid models replicate the complex environment of the human inner ear. These models may offer a more representative system for gene delivery and drug screening [120]. In the future, testing via large animal models will also be essential to mimic human cochlear anatomy and immune response.

In summary, therapeutics for *GJB2*-related deafness are poised to emerge in the near future. The combinations of AAV vectors, innovative base editing systems, and scalable modeling systems represent the path toward treatment of the most common cause of hereditary deafness globally through groundbreaking technological advancements.

## 8. Conclusions

Overall, GJB2 hearing loss remains the most common cause of genetic HL worldwide, with over 400 variants and a broad phenotypic spectrum spanning from late onset mild impairment to profound congenital deafness. Studies concerning the natural history of DFNB1 continue to reveal new insights into onset, progression, and auditory outcomes and how these outcomes may correlate with specific genotypes. Although most cases present as congenital and stable, there is a significant portion of cases demonstrating progressive or delayed onset HL, highlighting the importance of audiologic monitoring and early genetic testing in hearing loss patients.

Genotype remains a strong determinant of phenotype, with truncating variants producing more severe HL than non-truncating variants; however, modifier genes and regulatory elements may further influence the phenotype based on more recent reports. Variant prevalence among population groups should be considered in genetic testing and patient counseling, as variants in both the coding and non-coding regions of GJB2 differ in specific regions. This variability emphasizes the need for testing strategies tailored to population-specific variant profiles to further improve the accuracy of diagnosis and genetic counseling.

Recent advances in molecular therapeutics have presented new and innovative avenues for the correction of GJB2-related hearing loss. Synthetic AAV vectors in addition to allele-specific suppression and base editing, have been shown to be safe and effective in vivo studies and are promising avenues for future pre-clinical studies. Alongside improvements in modeling the inner ear, via inner ear organoids and large animal models, we are closer than ever to the first human gene therapy trials.

Together, these scientific and clinical advancements represent a paradigm shift from supportive to disease-modifying precision therapy for GJB2-related deafness. Integrating these scientific advancements with newborn hearing screening, early genetic testing, and precise intervention will be key to achieving the best outcomes for GJB2 patients. Continued collaboration across molecular genetics, auditory neuroscience, and clinical otology will be essential to translate these discoveries into accessible cures for those with GJB2 hearing loss.

## Data Availability

No new data were created or analyzed in this study. Data sharing is not applicable to this article.

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
