# Peer review of "GJB2-Related Hearing Loss: Genotype-Phenotype Correlations, Natural History, and Emerging Therapeutic Strategies"

_ijms, 2026, doi:10.3390/ijms27010491_

Round 1
Reviewer 1 Report
Comments and Suggestions for Authors
This is an interesting review about the molecular mechanisms, natural history, and emerging therapeutic strategies concerning GJB2-related hearing loss. This article is well written and provide an updated and comprehensive perspective of GJB2-related deafness and its emergent treatment landscape, including cochlear implantation in patients, as well as gene therapy and gene editing approaches in animal models. For me, this article is acceptable in the present form.
Author Response
Comment 1: This is an interesting review about the molecular mechanisms, natural history, and emerging therapeutic strategies concerning GJB2-related hearing loss. This article is well written and provide an updated and comprehensive perspective of GJB2-related deafness and its emergent treatment landscape, including cochlear implantation in patients, as well as gene therapy and gene editing approaches in animal models. For me, this article is acceptable in the present form. 
Response 1: We thank reviewer one for the positive assessment of our review manuscript and for finding it acceptable in its current form.
Reviewer 2 Report
Comments and Suggestions for Authors
A gap junction is a channel that mediates direct cell-to-cell transport of ions and other substances. A pair of hemichannels forms it, each comprised of six connexin proteins. Connexin 26 (Cx26), encoded by GJB2, is one of the connexin proteins. The physiological importance and the role of GJB2 for hearing have been experimentally demonstrated using mouse models. Consistently, a large number of GJB2 variants are identified in human patients suffering from hearing loss. This Review article by Morris et al. overviews the current status and recent advancements in research on GJB2-related hearing loss. Below, I list multiple points that I would like the authors to address.
Specific points
- The title does not accurately reflect the content of the manuscript, as "molecular mechanisms" are barely mentioned. Also, I think it would be better to say “GJB2-related hearing loss”, as the authors mention both recessively and dominantly (DFNA3A) inherited GJB2 variants.
- The physiological role of GJB2 is only minimally mentioned in a short paragraph under the Introduction section (lines 61-68). Please provide more thorough background information on the function and physiological role of GJB2 in hearing; otherwise, non-expert readers cannot fully appreciate the pathogenic significance of GJB2 variants.
- Related to the above point, I also suggest referring to the Gjb2 mouse models generated to date. These animal models significantly contributed to our understanding of the physiological role of Cx26 in mammalian hearing and served as in vivo platforms for testing gene therapies.
- Figure 1 (bottom panel), which provides only a linear schematic, is insufficient for appreciating how missense variants disrupt the function of Cx26. Studies that defined the Cx26 protein structure should be cited for readers who are interested in evaluating the functional impacts of missense variants within the protein's tertiary structure.
- Lines 184-185. Please be specific about “a single pathogenic allele”. Truncating GJB2 variants? If so, is this argument based on the assumption that all truncating GJB2 variants should result in null (and thus should be inherited recessively)?
Other points
- The physiological role of GJB2 in the cochlea is considered established. It is awkward to cite a preprint (Ref. #9) for introducing the established knowledge (line 65).
- Table 1 and Figures 1 and 2 are not referred to in the text.
- The abbreviations in Fig. 1 need to be explained in the figure legend (e.g., E1, extracellular loop 1). In line 164, the authors say “first extracellular loop of GJB2 (EC1)”. Please ensure that a consistent abbreviation is used throughout the manuscript.
- Line 375. “the large number of pathogenic variants” shouldn’t be an issue for a replacement therapy if most of them are recessively inherited.
- Line 411. The strategy used in Ref #115 is gene editing and is considered permanent.
